# Self-Assembly of Single-Polymer-Tethered Nanoparticle Amphiphiles upon Varying Tail Length

**DOI:** 10.3390/nano10112108

**Published:** 2020-10-23

**Authors:** Qingxiao Li, You-Liang Zhu, Xinhui Zhang, Kaidong Xu, Jina Wang, Zhixin Li, Yun Bao

**Affiliations:** 1School of Material and Chemical Engineering, Henan University of Urban Construction, Pingdingshan 467036, China; zhangxinhui@hncj.edu.cn (X.Z.); kdxu@hncj.edu.cn (K.X.); wangjina@hncj.edu.cn (J.W.); li.zhixin1989@163.com (Z.L.); 20181020@hncj.edu.cn (Y.B.); 2State Key Laboratory of Polymer Physics and Chemistry, Changchun Institute of Applied Chemistry, Chinese Academy of Sciences, Changchun 130022, China

**Keywords:** self-assembly, single-polymer-tethered nanoparticle amphiphiles, varying tail length, Brownian dynamics simulations

## Abstract

We systematically investigated the roles of tail length on the self-assembly of shape amphiphiles composed of a hydrophobic polymer chain (tail) and a hydrophilic nanoparticle in selective solvent using Brownian dynamics simulations. The shape amphiphiles exhibited a variety of self-assembled aggregate morphologies which can be tuned by changing tail length (n) in combination with amphiphile concentration (φ) and system temperature ($T^{*}$). Specifically, at high φ with $T^{*}=1.4$, the morphology varied following the sequence “spheres → cylinders → vesicles” upon increasing n, agreeing well with experimental observations. At low φ with $T^{*}=1.4$ or at high φ with $T^{*}=1.2$, the morphology sequence becomes “spheres or spheres and cylinders mixture → cylinders → vesicles → spheres” upon increasing n, which has not been found experimentally. Two morphological phase diagrams depending on n and φ were constructed for $T^{*}=1.4$ and 1.2, respectively. The rich phase behaviors on varying tail length could provide the feasible routes to fabricate target aggregate morphologies in various applications, especially for the vesicles with tunable thickness of membranes that are crucial in drug and gene delivery.

## 1. Introduction

The distinct properties of two blocks in a molecule including shape [1,2,3,4,5,6,7,8,9,10,11,12,13,14,15,16,17,18,19,20,21,22,23], compatibility [24,25,26], rigidity [27] and charge [28,29] offer many opportunities in controlling the morphologies by micro-phase separation. Of these, understanding the self-assembly of amphiphilic polymer-tethered nanoparticles is critical to developing novel fabrication materials with highly diverse and thermodynamically stable and metastable structures in the solution [1,2,3,4,5,6,7,8,9,10], bulk [4,5] and thin-film [4,5]. The shape of the nanoparticles with certain rigidity and anisotropy has been recognized as an important parameter in the self-assembly of amphiphiles. “Shape amphiphiles” thus refer to the entities constructed from those chemically distinct and geometrically anisotropic nano-building blocks [1], such as polystyrene-hydrophilic fullerene (C_60_) [1,2,3,4,5] conjugates, polystyrene-hydrophilic polyhedral oligomeric silsesquioxane (POSS) [3,4,5,6,7,8] conjugates, polymer-tethered inorganic/organic nanoparticles [9], poly(2-(dimethylamino)ethyl methacrylate)-block-polystyrene [10], etc. The studies on their self-assembled behaviors are not only scientifically intriguing, but also technologically anticipating in nanopatterning fabrications and others [5].

In dilute solutions, the self-assembled morphologies of amphiphilic block copolymers have been identified, and the mechanisms of morphology formation and transitions have been elucidated [30,31,32,33,34,35,36,37,38,39]. In contrast, the self-assembly of shape amphiphiles is much less understood than that of amphiphilic block copolymers. Therefore, computer simulations have been employed to give insight into the self-assembly of shape amphiphiles. Specifically, Wang’s group investigated the self-assembly of shape amphiphiles in selective solvents by employing the dissipative particle dynamics approach [11,12]. They found that the shape amphiphiles tethered by one tail at the length of 5–31 beads (the size of a hydrophilic H-bead is 1.5–2.0 times of a tail bead in their study) can self-assemble into vesicles, worm-like cylinders, spheres, etc. by changing the interaction parameters between hydrophilic nanoparticle and solvent, the length of hydrophobic tail or the concentration of amphiphiles. Moreover, the amphiphiles with short tails tend to form vesicles, while the ones with long tails prefer to form spherical micelles [11]. In our previous work, we studied the self-assembly of giant amphiphiles based on polymer-tethered nanoparticle in selective solvents [13]. We found that the shape amphiphiles with one tail of 16 beads self-assembled into spherical micelles, while the shape amphiphiles with one tail of 48 beads self-assembled into cylindrical micelles at the reduced temperature of 1.2 and the amphiphile concentration of 3.2% (the diameter of a hydrophilic H-bead is three times that of a tail bead) [13]. These results present significant advances in the simulation studies of the self-assembly of the polymer-tethered nanoparticle shape amphiphiles with short polymer tails. However, for the shape amphiphiles with a much longer polymer tail, computer simulations have not been explored to understand their self-assembly in dilute solution.

Recently, Cheng’s group designed and synthesized shape amphiphiles with a much longer polymer tail based on carboxylic acid-functionalized fullerene (AC_60_) [1] or carboxylic acid-functionalized polyhedral oligomeric silsesquioxane (APOSS) [6] as hydrophilic nanoparticle and polystyrene (PS) as hydrophobic polymer tail. In a common solvent, which was a mixture of 1, 4-dioxane and dimethylformamide (DMF) (*w*/*w* = 1/1) with water serving as selective solvent, self-assembly of the series of PS_n_–AC_60_ has been systematically studied by varying hydrophobic tail length and/or amphiphilic concentration [1]. They found that the self-assembly morphologies of PS_n_–AC_60_ amphiphiles were always the spherical micelles in the low amphiphilic concentration range of equal to or less than 0.25 (wt)%; and the morphological transitions of aggregates occur from spherical micelles to cylindrical micelles and further to bilayer vesicles as increasing the amphiphilic concentration 0.25 (wt)% and/or the length of PS tail [1]. They also found that PS_n_–APOSS in selective solvents can self-assemble into vesicles, worm-like cylinders and spheres upon increasing the degree of ionization of the carboxylic acid groups on the POSS nanoparticles [6]. These findings indicate that the self-assembly of the shape amphiphiles as a unique class of new materials arrests the essential structural features of the traditional amphiphilic block copolymers, but are not completely the same as copolymers due to the nanoparticle possessing much larger sizes [4,5,6]. Therefore, there are many issues remaining, such as the effects of the coordination of the much longer hydrophobic tail and the amphiphile concentration at different temperatures on the morphologies and the dynamic formation processes of aggregates. Furthermore, the pathways of vesicle formation of shape amphiphiles with a longer tail are still unclear. There are two pathways (Mechanisms I and II) of vesicle formation in amphiphilic block copolymers that have been found and corroborated [40,41,42,43,44,45,46,47,48,49]. In Mechanism I, vesicles are formed via intermediated bilayer-type structures [40,41,42,43,44,45,46]. In Mechanism II, the intermediated structures are large micelles [44,45,46,47,48,49]. In the systems of polymer-tethered nanoparticle shape amphiphiles with two polymer tails, the transitions between Mechanisms I and II happened by controlling the hydrophobic tail length or the amphiphile concentration [13]. However, the effect of the tail length on the pathways of vesicle formation has not been exposed systematically before.

Brownian dynamics as a stochastic molecular dynamics simulation method allows the simulations of long time and large systems [14,15,16]. As shown by Glotzer and coworkers [14,15,16,17,18,19,20,21,22,23], Brownian dynamics is a very promising approach to investigate the self-assembly behaviors of the polymer-tethered nanoparticle shape amphiphiles [16]. Therefore, the primary objective of this study was, using Brownian dynamics method, to detect the phase behaviors and the dynamic formation processes of typical aggregates from the shape amphiphiles with one polymer tail in dilute solution. Unlike hydrophobic tail length n=16 or 48 used in our previous works [13], we performed the simulations in a much wide range of tail length. Specifically, the conditions of $T^{*}=1.4$ and n=70−140 as well as $T^{*}=1.2$ and n=54−120 were investigated. Two morphological phase diagrams depending on n and φ were constructed for $T^{*}=1.4$ and 1.2, respectively. The pathways of aggregate formation were distinguished for different tail lengths. Importantly, the thickness of membranes of vesicles that is crucial in the release of drug and gene was tuned by altering the tail length. Moreover, the relationship between the membrane thickness and the tail length was revealed. The reason for the formation of vesicles or micelles at different tail length was explained. These simulation results not only expose the roles of the tail length in the self-assembly of shape amphiphile, but also could be exploited in the future fabrication of aggregated structures.

## 2. Simulation and Method

Our study focused on the self-assembly of polymer-tethered nanoparticle shape amphiphiles in a selective solvent using the Brownian dynamics approach. Each model amphiphile is composed of a linear hydrophobic polymer tail (P_n_) and a hydrophilic nanoparticle (H-bead), denoted as P_n_–H, as shown in Figure 1. The diameter of the nanoparticle is three times that of a polymer tail bead, the n in the denotation of model amphiphile is the number of coarse-grained beads (P-bead) on the hydrophobic tail, and its value was in the range of 23–180 in our simulations, which is similar to the shape amphiphiles synthesized by Yu et al. [1].

The non-bonded P–P, P–H, and H–H interactions in coarse-grained beads are represented by a radically modified and Shifted Lennard-Jones (SLJ) potential [13],
(1)$$V_{m-\mathrm{SLJ}}\left(r_{ij}\right)=\{\begin{array}{ll} 4\varepsilon_{ij}[(\frac{\sigma }{r_{ij}-\Delta_{ij}}{)}^{12}-\alpha_{ij}(\frac{\sigma }{r_{ij}-\Delta_{ij}}{)}^{6}]-4\varepsilon_{ij}[(\frac{\sigma }{\beta_{ij}}{)}^{12}-\alpha_{ij}(\frac{\sigma }{\beta_{ij}}{)}^{6}] & r_{ij}\leq \beta_{ij}-\Delta_{ij}\\ 0 & r_{ij}\leq \beta_{ij}-\Delta_{ij} \end{array}$$
where $\varepsilon_{ij}$ is the interaction potential well depth, σ is the diameter of a P-bead, $r_{ij}$ is the distance between two beads, $\beta_{ij}$ is the cutoff radius and $\Delta_{ij}=\frac{d_{i}+d_{j}}{2}-\sigma$ is the shift distance, with $d_{i}$ and $d_{j}$ the diameter of *i*th and *j*th beads, respectively. The parameter $\alpha_{ij}$ is a factor between 0 and 1.0 to reflect the hydrophilicity or hydrophobicity of the beads [13,50,51]. In $\varepsilon_{ij}$, $\beta_{ij}$, $\Delta_{ij}$, $d_{i}$, $d_{j}$ and $\alpha_{ij}$, the subscripts *i* and *j* refer to P and H for P-bead and H-bead, respectively.

The bonding interactions between neighboring coarse-grained beads along the tail of the shape amphiphiles is given by the finitely extensible nonlinear elastic (FENE) springs [13,22,23,52]
(2)$$V\left(r_{ij}\right)=\{\begin{array}{ll} -\frac{1}{2}kR_{0}^{2}ln[1-\left(\frac{r_{ij-\Delta_{ij}}}{R_{0}}{)}^{2}\right] & r_{ij}\leq R_{0}\\ 0 & r_{ij}>R_{0} \end{array}$$
where k is the FENE spring constant and $R_{0}$ is the maximum bond length. Here, we set $k=30.0\varepsilon /\sigma^{2}$ and $R_{0}=1.5\sigma$.

The Brownian dynamics simulation method was proven to be suitable for research into the self-assembly of polymer-tethered nanoparticle shape amphiphiles [13,14,15,16,17,18,19,20,21,22,23]. In Brownian dynamics, the motion of each bead in the system is subjected to conservative, random and frictional forces $F_{i}^{c}\left(r_{i}\left(t\right)\right)$, $F_{i}^{R}\left(t\right)$ and $\gamma_{i}v_{i}\left(t\right)$, respectively, and follows the Langevin equation:(3)$$m_{i}{\ddot{r}}_{i}\left(t\right)=F_{i}^{C}\left(r_{i}\left(t\right)\right)+F_{i}^{R}\left(t\right)-\gamma_{i}v_{i}\left(t\right)$$
where $m_{i}$, $r_{i}$, $v_{i}$ and $\gamma_{i}$ are the mass, position, velocity and friction coefficient of the *i*th bead, respectively. *t* is the time. Here, we set $\gamma_{P}=1.0$ for P-bead and $\gamma_{H}=3.0$ for H-bead. The solvent effects are implicitly adopted by the random force that satisfies the fluctuation–dissipation theorem:(4)$$\langle F_{i}^{R}\left(t\right)F_{j}^{R}\left(t^{\prime }\right)\rangle =6\gamma_{i}k_{B}T\delta_{ij}\left(t-t^{\prime }\right)$$
where $k_{B}$ is the Boltzmann constant and T is the temperature. The coupling between friction and random forces acts as an effective thermostat [13,53].

All simulations were performed in an NVT ensemble in a cubic box under the periodic boundary conditions. The box side length was set at 100. The concentrations of the amphiphiles (φ) and hydrophilic beads ($\varphi_{H}$) in the cubic simulation box are defined as $\varphi =\frac{N_{H}V_{H}+N_{P}V_{P}}{V}$ and $\varphi_{H}=\frac{N_{H}V_{H}}{V}$, respectively, in which $N_{H}$ and $N_{P}$ are the total numbers of hydrophilic H-beads and hydrophobic P-beads, respectively. $V_{H}$, $V_{P}$ and V are the volume of H-bead, P-bead and the cubic simulation box, respectively. $\varepsilon_{PP}=1.0$, $\sigma =d_{P}=1.0$ and the mass of a P bead $m_{P}=1.0$ are taken as the basic units of energy, length and mass, respectively, thus the time unit $\tau =\sigma \sqrt{m_{P}/\varepsilon_{PP}}$. We fixed the mass of an H-bead at M=20.0, the diameter of an H-bead at $d_{H}=3.0$, the cutoff radius at $\beta_{PP}=3.0$, $\beta_{HH}=2^{\frac{1}{6}}$ and change parameters $\alpha_{PP}$ and $\alpha_{HH}$ in the ranges of 0.8–1.0 and 0–0.5 to reflect hydrophobicity of the P-beads and hydrophilicity of the H-beads, respectively.$\;\varepsilon_{HH}$, $\beta_{\mathrm{PH}}$ and $\varepsilon_{\mathrm{PH}}$ were also changed to reproduce the available experimental results. The amphiphile concentration and the reduced temperature ($T^{*}=k_{B}T/\varepsilon_{PP}$) were varied in the ranges of 0.8–6.2% and 1.2–1.4, respectively, to investigate their effects on the self-assembly behaviors of the shape amphiphiles. We checked the size effect of the simulation box, which is larger than simulation systems with box side length of 140 or 200 with φ=3.2% at $T^{*}=1.4$, and found that good reproducibility was obtained. An integration time step of δt=0.005 was used, and the simulations lasted $2.0\times 10^{8}\delta t$ or $4.0\times 10^{8}\delta t$. All simulations were performed with HOOMD package [54,55,56] and some data with GALAMOST [57] on an NVIDIA GeForce GTX 1080Ti GPU.

## 3. Results and Discussion

In this study, we investigated the roles of tail length on the formation and transitions of morphology of shape amphiphiles consisting of a hydrophilic nanoparticle bead and a grafted hydrophobic polymer tail (named as P_n_–H) in a selective solvent by changing tail length n at certain amphiphile concentration φ and system temperature $T^{*}$. The effects of these parameters on the self-assembled morphologies of the shape amphiphiles were studied by testing possible parameters. Figure 2a shows the morphological phase diagram of P_n_–H as a function of n and φ at $T^{*}=1.4$. Typical snapshots are shown in Figure 2b. We found that the morphologies of the aggregates are significantly influenced by φ and n. Specifically, five distinct types of aggregates are observed: spherical micelles, worm-like cylinders, cylindrical networks, bilayer vesicles and tubular vesicles. At the low concentration range of φ≤1.4%, the P_n_–H amphiphiles with hydrophobic tail length in the range of n=23–140 form spherical micelles, and the size of micelles increases upon increasing n. When n and φ are relatively larger, the transitions of the self-assembled morphologies depending on φ or n are given as follows: for P_46_–H amphiphiles, the self-assembled morphology changes from spheres at φ=1.4%, to worm-like aggregates in the range of φ=1.7%–2.6% and finally to cylindrical networks in the range of φ=3.2%–6.2%; and for P_70_–H amphiphiles, the spherical aggregates remain until φ=2.0%, then change to worm-like aggregates at φ=2.6%, further to spherical vesicles in the range of φ=3.2%–5.6% and finally to tubular vesicles at φ=6.2%. However, for n=85–140, the morphology of aggregates transforms from spherical micelles directly to spherical vesicles upon increasing φ, without worm-like aggregates. The critical amphiphile concentration ($\varphi_{c}$) of the vesicle formation is the lowest at n=85 (φ=2.6%) and slightly increases upon increasing n.

Figure 3a,b shows the morphological phase diagram and the typical snapshots of the aggregates formed by the P_n_–H amphiphiles as a function of φ and n at $T^{*}=1.2$. It is clearly shown that the self-assembled morphologies of the P_n_–H systems are spherical micelles in the lower amphiphile concentration range of φ≤0.8%. With the increase of φ, at n=23, the self-assembled morphology is a mixture of spherical micelles and worm-like cylinders in the range of φ=1.4%–6.2%; at n=46, the self-assembled morphology changes from a mixture of spherical micelles and worm-like cylinders in the range of φ=1.4%–3.2% to worm-like cylinders in the range of φ=3.8%–4.4% and further to Y-like cylinders in the range of φ=5.0%–6.2%; at n=54, the self-assembled morphology changes from spherical micelles at φ=1.4% to mixed morphologies of spherical micelles and worm-like cylinders in the range of φ=2.0%–3.2%, further to worm-like cylinders in the range of φ=3.8%–4.4% and finally to bilayer vesicles in the range of φ=5.0%–6.2%; at n=56–120, the morphology transition occurs from spherical micelles directly to bilayer vesicles, and the transitions are similar to the above simulation results at $T^{*}=1.4$; and at n=140, the self-assembled morphology is only single spherical micelles. The $\varphi_{c}$ value of the vesicle formation is the lowest at n=56 (φ=2.6%); it also slightly increases with the increasing of n, which is similar to that of $T^{*}=1.4$. We observed cylindrical micelles for the shape amphiphiles with a short tail (n < 80 for $T^{*}=1.4$ and n < 55 for $T^{*}=1.2$), while we did not observe any cylindrical micelles for the shape amphiphiles with a longer tail. The shape amphiphiles with a longer tail have relatively fewer shape heads per volume of tail, but a larger surface area needs more hydrophilic heads to minimize the surficial free energies. Thereby, it is difficult to form cylindrical micelles for the shape amphiphiles with a long tail.

A comparison of Figure 2 and Figure 3 shows that the two morphological phase diagrams are quite different from each other, indicating that the temperature $T^{*}$ has an important effect on the self-assembled morphology of aggregates. One difference is that systems with n=23–54 can form mixed morphologies of spheres and cylinders in relatively wide range of *φ* at $T^{*}=1.2$, while mixed morphologies of spheres and cylinders at n=23–140 and $T^{*}=1.4$ are not found. Another difference is that, compared with the phase diagram at $T^{*}=1.4$, the windows of the vesicles and cylinders in phase diagram at $T^{*}=1.2$ shift to the directions of smaller n and higher *φ*. The third difference is that, in the phase diagram at $T^{*}=1.2$, the morphology of aggregates is only spherical micelles at n=140, while, in the phase diagram at $T^{*}=1.4$ and n=140, the morphology of aggregates changes from spherical micelles to vesicles upon increasing φ.

The above simulation results are comparable to previous experiments. Upon increasing *φ*, the transitions of the aggregate morphologies occur following the sequence “spherical micelles → worm-like cylinders → spherical vesicles → tubular vesicles” at n=70 and $T^{*}=1.4$ and “spheres → mixed morphologies of spheres and cylinders → worm-like cylinders → vesicles” at n=54 and $T^{*}=1.2$, which are similar to that observed experimentally by Yu et al. [1]. In their experiments, the self-assembled morphology of PS_70_–AC_60_ in mixture of 1,4-dioxane/DMF/water solution changed from spherical micelles with an initial molecular concentration of 0.1 (wt)%, to cylinder networks at 0.5 (wt)%, to a mixed morphology of cylinders and vesicles at 1.0 (wt)% and finally to vesicles at 2.0 (wt)%.Similar aggregate morphologies were also observed from PS_100_–AC_60_ in their experiments. Furthermore, their experiments also indicated that the window for worm-like cylinders became narrower as increasing the tail length of PS, which is consistent with our result. The morphology transitions of the P_n_–H self-assemblies are also similar to the experimental findings of amphiphilic block copolymers by Eisenberg and co-workers [30,31]. In their experiments, the aggregate morphology of polystyrene_190_-b-poly(acrylic acid)_20_ in DMF–water mixtures changed from spheres to rods and vesicles as the copolymer concentration was increased from 1 to 3.5(wt)% [30,31]. For much longer hydrophobic tails (n=85–140 and n=56–120 in Figure 2 and Figure 3, respectively), the aggregate morphologies transform from spherical micelles directly to bilayer vesicles with the increasing of *φ*, and the $\varphi_{c}$ value increases slightly with increasing n, which are different from that observed experimentally by Yu et al. [1]. On the other hand, the morphological transitions of aggregates from spherical micelles to cylindrical networks and further to bilayer vesicles are observed upon increasing n (4.4%≤φ≤6.2% in Figure 2). The simulation results are in good agreement with experimental findings [1]. For example, in the high molecular concentration range between 1.5 (wt)% and 2.0 (wt)%, the morphology transitions of aggregates appear from spherical micelles for PS_23_–AC_60_, to cylinders for PS_46_–AC_60_ and further to vesicles for PS_70_–AC_60_ [1]. The results are also consistent with experimental findings by Eisenberg and co-workers [30]. For example, by increasing hydrophobic/hydrophilic block length ratio ($n_{l}$), in DMF–water mixtures with a molecule concentration of 2.0 (wt)%, the aggregate morphological transition of polystyrene_200_-b-poly(acrylic acid)_n_ changed from spheres at $n_{l}=200/21$ to rods at $n_{l}=200/15$ and further to vesicles at $n_{l}=200/8$ [30]. However, upon increasing n, the morphology transitions of aggregates appeared from spherical micelles or mixed morphologies of spheres and cylinders, to worm-like cylinders or Y-like cylinders, further to bilayer vesicles and finally again to spheres (2.6%≤φ≤3.8% and 3.8%≤φ≤6.2% in Figure 2 and Figure 3, respectively); and from mixed morphologies of spheres and cylinders to vesicles and further to spheres (2.6%≤φ≤3.2% in Figure 3). These simulation results are not observed experimentally.

Figure 4 summarizes the formation processes and dominant morphologies obtained at different time steps and from (Figure 4c) different P_n_–H amphiphiles in a selective solvent at φ=4.4% and $T^{*}=1.4$; and (Figure 4d) different φ for P_70_–H amphiphiles at $T^{*}=1.4$. It is noted that, with the increase of the tethered polymer length, or with the decrease of the amphiphilic concentration, the formation pathway of vesicles changes form Mechanism I to Mechanism II. The simulation results are consistent with that obtained from the polymer-tethered nanoparticle amphiphiles with two tails in a selective solvent [13]. The simulation results are also consistent with dissipative particle dynamics simulation for amphiphilic triblock copolymer solution systems where a transition of the vesicle formation pathway from Mechanism I to Mechanism II was found while increasing the hydrophobic block length or decreasing the amphiphilic concentration [44].

We analyzed the structural characteristics of vesicle formation at different hydrophobic tail length n and certain amphiphile concentration φ. As shown in Figure 5, the outer radius of the vesicle is denoted as $R_{out}$, the inner radius is denoted as $R_{in}$ and the wall thickness is denoted as d, which is determined by the difference between the outer and inner hydrophobic membranes. Here, the outer or inner radius is defined as the average distance from the center of mass of the vesicle to the surface [58]. The wall thickness d is the average thickness of hydrophobic membrane. The changes of the $R_{out}$, d and $R_{in}$ were therefore analyzed for vesicle characteristics based on different n and φ.

The volume of the membrane of the vesicle filled by the tails of P_n_–H is denoted as $V^{\prime }$. The surface area of the vesicle consisting of inner one and outer one formed by the heads of P_n_–H is denoted as $S^{\prime }$. The relationships for $R_{out}$, $R_{in}$, $S^{\prime }$ and $V^{\prime }$ are given in Equations (5) and (6):(5)$$R_{out}^{2}+R_{in}^{2}=S^{\prime }/4\pi$$
(6)$$R_{out}^{3}-R_{in}^{3}=3V^{\prime }/4\pi$$

The volume $V^{\prime }$ could be simply calculated by the occupied volume of one tail and the number of tails. The surface area $S^{\prime }$ could be ideally calculated by the number and size of head nanoparticles under the hypothesis of that the particles are closely packed at surface to minimize surface tension. From Equation (6), we can learn $R_{out}\geq R_{in}$. When $R_{out}=R_{in}$, the wall thickness of the hydrophobic membrane d is 0, namely the hydrophobic tail length n is 0. When φ is fixed, upon increasing n, the volume of the hydrophobic membrane increases. Meanwhile, the outer radius $R_{out}$ is increased and the inner radius $R_{in}$ is decreased at a fixed surface area. Thereby, the thickness of the membrane of vesicles becomes increased upon increasing n. When n increases to a critical value, the inner radius $R_{in}$ becomes 0. In this case, the aggregate morphology becomes spherical micelle. For larger n, the aggregates are still spherical micelles due to the minimization of surface free energies. The speculations are confirmed by the parameters given in Table 1 and the cross-sectional slices shown in Figure 6a of the aggregates self-assembled from P_70_–H, P_85_–H, P_100_–H, P_120_–H, P_140_–H and P_180_–H when the amphiphile concentration is φ=4.4%. In Table 1 and Figure 6a, we can see that the inner radius $R_{in}$ decreases with the increase of n and the aggregate morphology is spherical micelles at n=180. The change of the aggregate morphology from vesicles into spherical gradually upon increasing n is consistent with the previous theoretical analysis. In Table 2 and Figure 6b, we can see that the inner radius $R_{in}$ and the outer radius $R_{out}$ of vesicle are both increased with the increase of amphiphile concentration φ at a fixed tail length n=100. However, the variation of the wall thickness of the hydrophobic membrane d is less significant. These simulation results help to design vesicles with controllable thickness, which are crucial in drug and gene delivery.

## 4. Conclusions

Brownian dynamics simulation with implicit solvent was used to study the self-assembly of shape amphiphiles composed of a hydrophobic polymer tail and a hydrophilic nanoparticle in a selective solvent. The shape amphiphiles are similar to PS–AC_60_ [1] or PS–APOSS [6], which can self-assemble into a wide variety of morphological structures. Dependent on hydrophobic tail length (n), the self-assembled aggregate morphologies at certain amphiphile concentration (φ) and system temperature ($T^{*}$) comprise seven types: spherical micelles, the mixed morphologies of spheres and cylinders, worm-like cylinders, cylindrical networks, Y-like cylinders, spherical vesicles and tubular vesicles. Two morphological phase diagrams as a function of n and φ were constructed at $T^{*}=1.4$ and 1.2, respectively. The transitions of aggregate morphology from spherical micelles to cylindrical networks and further to bilayer vesicles are observed by increasing n at 4.4%≤φ≤6.2% and $T^{*}=1.4$. The simulation results are in good agreement with experimental findings of Yu et al. [1]. Further, more relationships about morphology aggregates that have not been found in experiments were exposed. For example, the transition appearing from spherical micelles or mixed morphologies of spheres and cylinders, to worm-like cylinders or Y-like cylinders, further to bilayer vesicles and finally again to spheres upon increasing n at 2.6%≤φ≤3.8% and $T^{*}=1.4$ or at 3.8%≤φ≤6.2% and $T^{*}=1.2$, as well as the transition from mixed morphologies of spheres and cylinders to vesicles and further to spheres at 2.6%≤φ≤3.2% and $T^{*}=1.2$ are presented. For a much longer polymer tail (n=85–140 in $T^{*}=1.4$ systems or n=56–120 in $T^{*}=1.2$ systems), the morphology transitions of aggregates occurring from spherical micelles directly to bilayer vesicles upon increasing of φ have also not been observed experimentally.

Our simulation work not only well complements experimental studies, but is also enlightening for material design that depends on aggregate morphologies. In addition to the systematical investigation of the aggregate morphologies that are controlled by thermodynamics and useful in diverse applications of shape amphiphiles upon varying tail length, we scrutinized the formation pathways that depend on dynamics and are difficult to be observed in experiments. A summary diagram of formation pathway depending on tail length and concentration is presented. More importantly, the thickness of membranes of vesicles, which is crucial in drug and gene delivery, was tuned by changing tail length. In addition, the relationship between the membrane thickness and tail length is given. The formation of vesicles or micelles on increasing tail length is explained. Therefore, our simulation work might be valuable for guiding experimental studies to manipulate the aggregation structures of polymer-tethered nanoparticle shape amphiphiles.

## Figures and Tables

**Figure 1 nanomaterials-10-02108-f001:**
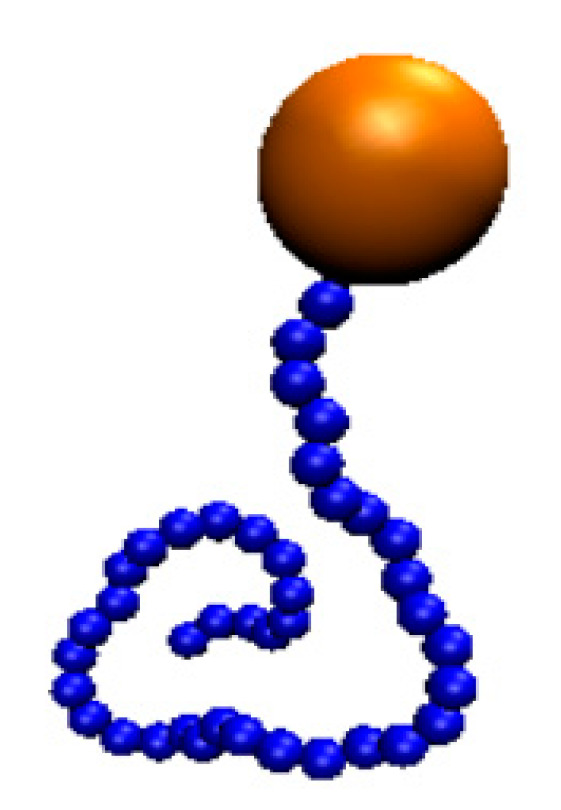
Model of a shape amphiphile (P_n_–H), in which the blue beads and the orange bead represents the hydrophobic polymer tail (P_n_) and the hydrophilic nanoparticle (H), respectively.

**Figure 2 nanomaterials-10-02108-f002:**
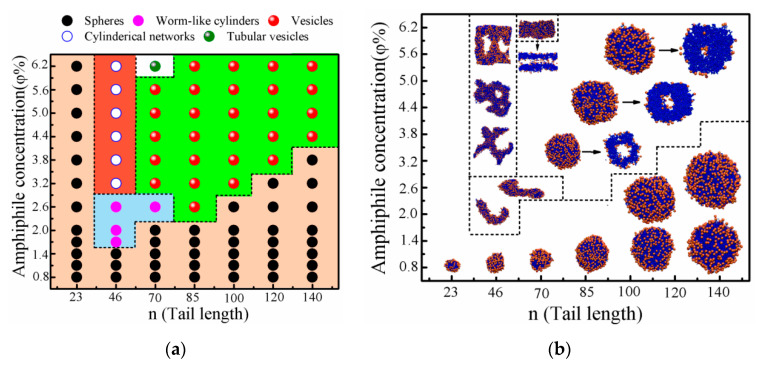
(**a**) Morphological phase diagram; and (**b**) typical snapshots of aggregates formed by the P_n_–H amphiphiles depending on φ and n at $T^{*}=1.4$. The color scheme is the same as that in Figure 1. Each arrow points to the cross-section slice of the corresponding aggregate.

**Figure 3 nanomaterials-10-02108-f003:**
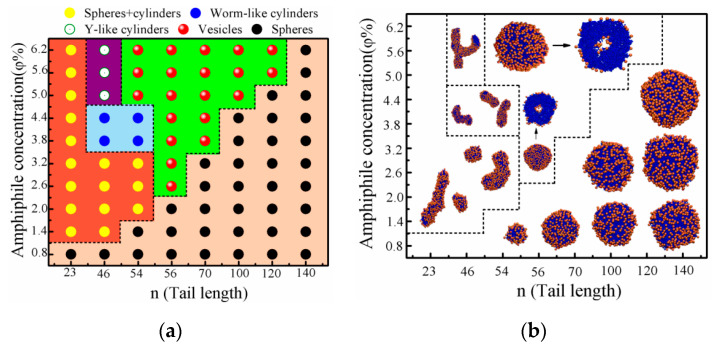
(**a**) Morphological phase diagram; and (**b**) typical snapshots of aggregates formed by the P_n_–H amphiphiles depending on φ and n at $T^{*}=1.2$. The color scheme and the meaning of arrows are the same as that in Figure 2.

**Figure 4 nanomaterials-10-02108-f004:**
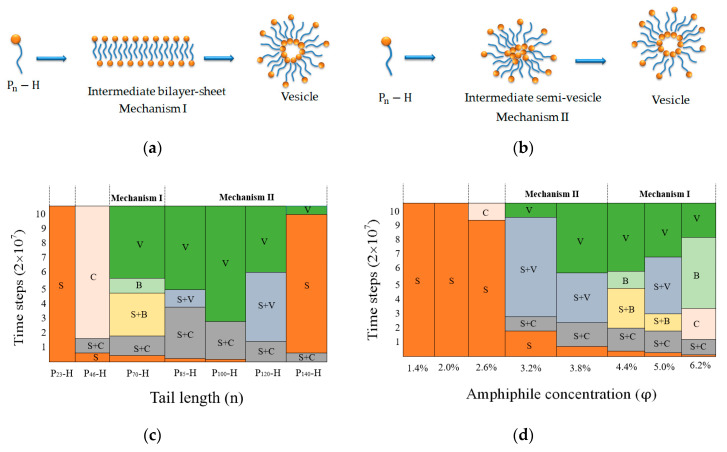
(**a**,**b**) Schematic drawings of Mechanisms I and II of the vesicle formation from P_n_–H amphiphiles, respectively. Dominant morphologies obtained at different time steps: (**c**) from different P_n_–H amphiphiles with φ=4.4% and $T^{*}=1.4$; and (**d**) from different φ for P_70_–H amphiphiles with $T^{*}=1.4$. These morphologies included: spherical micelles (S), cylindrical micelles (C), bilayer sheet (B) and vesicles (V). Each datum is divided into zones based on its respective mechanism classification.

**Figure 5 nanomaterials-10-02108-f005:**
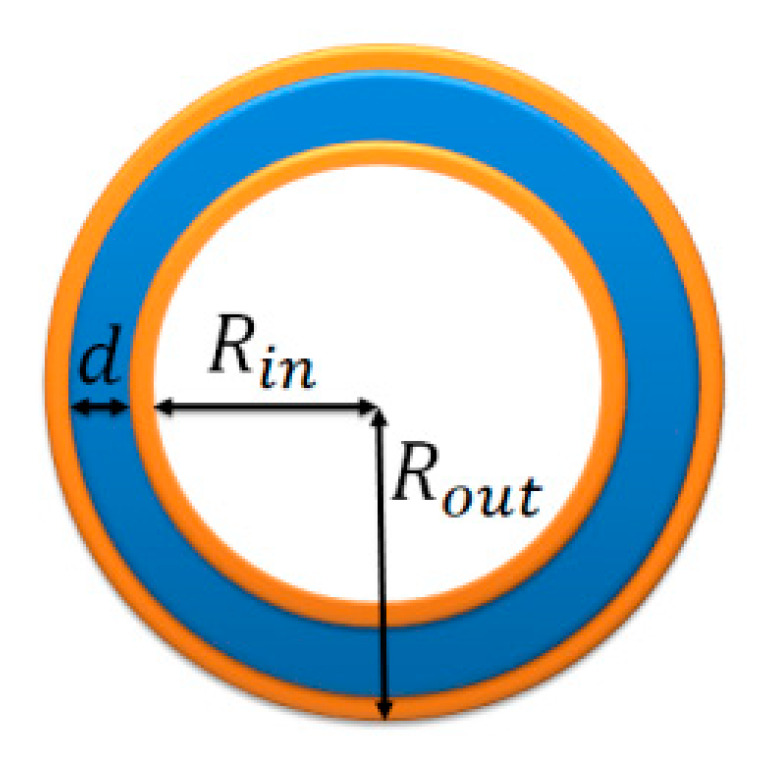
Schematic illustration of a typical vesicle, where $R_{in}$ is the inner radius, d is the wall thickness of the hydrophobic membrane and $R_{out}$ is the outer radius.

**Figure 6 nanomaterials-10-02108-f006:**
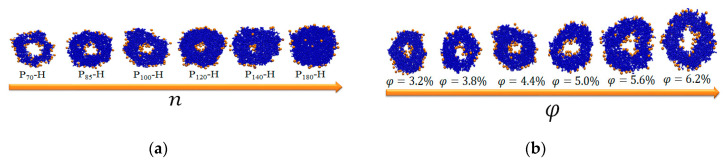
(**a**) Cross-sectional slices of the aggregates formed by P_70_–H, P_85_–H, P_100_–H, P_120_–H, P_140_–H and P_180_–H and the amphiphile concentration φ=4.4%. (**b**) Cross-sectional slices of the aggregates formed by P_100_–H amphiphiles and different amphiphile concentration φ=3.2%, φ=3.8%, φ=4.4%, φ=5.0%, φ=5.6% and φ=6.2%. The orange and blue beads represent hydrophilic nanoparticle and hydrophobic polymer tail, respectively.

**Table 1 nanomaterials-10-02108-t001:** Summary of the inner and outer membrane radius and of the bilayer thickness with different tail length for P_n_–H with φ=4.4% at $T^{*}=1.4$.

n	N ^1^	$R_{in}$	$R_{out}$	d
70	867	15.02	35.42	20.40
85	751	14.20	35.74	21.54
100	662	12.17	36.38	24.21
120	572	8.71	36.71	28.00
140	504	3.32	36.73	33.41
180	406	0.00	37.17	37.17

^1^ N is molecule number.

**Table 2 nanomaterials-10-02108-t002:** Summary of the inner and outer membrane radius and of the bilayer thickness with different amphiphile concentration for P_100_–H at $T^{*}=1.4$.

φ	N ^1^	$R_{in}$	$R_{out}$	d
3.2%	482	7.05	31.49	24.44
3.8%	572	10.04	34.33	24.29
4.4%	662	12.17	36.38	24.21
5.0%	753	13.04	37.19	24.15
5.6%	843	14.92	39.02	24.10
6.2%	933	15.49	39.51	24.02

^1^ N is molecule number.

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
