# Peer review of "Self-Assembly of Single-Polymer-Tethered Nanoparticle Amphiphiles upon Varying Tail Length"

_nanomaterials, 2020, doi:10.3390/nano10112108_

Round 1

Reviewer 1 Report

This paper aims at investigating, through Brownian dynamics simulations, the phase behaviour and the mechanisms of formation of aggregates of shape amphiphiles with a polymer tail of increasing length and amphiphile concentration, at two different temperatures.

This paper uses the same approach that the authors used in ref 23, whilst pushes a little forward the investigation of the formation of shape amphiphile aggregates by focussing on a larger tail length range. Also, a new interpretation of the vesicle formation is given by the investigation of the thickness of the bilayer and the radius of the outer and inner membrane, respectively. This last part would merit to be deepened in order to give a sound of novelty to the paper compared to ref 23.

Specific comments:

  1. The authors should provide more quantitative data, both as inputs and outputs, of the simulations of the inner and outer membrane radius and of the bilayer thickness

The paper should be published after the minor revisions following the requirements here above.

Author Response

We thank the reviewer for the positive comments and helpful suggestion. We have added quantitative data including the inner and outer radius, and the bilayer thickness of membrane to improve the last part. Please see Table 1, Table 2, and the last paragraph of Results and Discussion section in revised manuscript.

Reviewer 2 Report

The authors report an application of Brownian dynamics to simulate the self-assembly of chain amphiphiles by varying their tail length, composition and temperature. The simulations were done mostly with HOOMD package that was developed at Glotzer Lab. NVIDIA GeForce GTX 1080Ti GPU with 3500 CUDA Cores was the hardware for these simulations.

The non-bonded interactions were modeled with shifted Lennard-Jones (SLJ) potentials, while the bonding interactions between neighboring coarse-grained beads were represented as nonlinear elastic springs.

In general the results are interesting and the article is well written. The introduction could be improved by adding a wider scope, or a bigger picture on the subject. There are a lot of investigations on the phase diagram of different chain polymers and lipids that form similar structures to the one reported by the authors. The introduction section will benefit if the following very central references to the subject of the article are properly mentioned:

Morphologies of block copolymers composed of charged and neutral blocks, Soft Matter, 2012,8, 3036-3052

Identification of large channels in cationic PEGylated cubosome nanoparticles, Soft Matter, 2015,11, 3686-3692

Author Response

We very much appreciate the reviewer for the kind and helpful comments. We have improved the introduction section by adding a bigger picture on the subject and properly citing the references suggested by the reviewer. Please see the first paragraph of the introduction in revised manuscript.